# CROSS-TASK KNOWLEDGE TRANSFER FOR VISUALLY-GROUNDED NAVIGATION

## ABSTRACT

Recent efforts on training visual navigation agents conditioned on language using deep reinforcement learning have been successful in learning policies for two different tasks: learning to follow navigational instructions and embodied question answering. In this paper, we aim to learn a multitask model capable of jointly learning both tasks, and transferring knowledge of words and their grounding in visual objects across tasks. The proposed model uses a novel Dual-Attention unit to disentangle the knowledge of words in the textual representations and visual objects in the visual representations, and align them with each other. This disentangled task-invariant alignment of representations facilitates grounding and knowledge transfer across both tasks. We show that the proposed model outperforms a range of baselines on both tasks in simulated 3D environments. We also show that this disentanglement of representations makes our model modular, interpretable, and allows for transfer to instructions containing new words by leveraging object detectors.

## 1 INTRODUCTION

Deep reinforcement learning has been shown to be capable of achieving super-human performance in playing games such as Atari 2600 (Mnih et al., 2013) and Go (Silver et al., 2016). Following the success of deep reinforcement learning in 3D Games such as Doom (Lample & Chaplot, 2017; Dosovitskiy & Koltun, 2017) and DeepmindLab (Mnih et al., 2016), there has been increased interest in using deep reinforcement learning for training *embodied* agents, which interact with a 3D environment by receiving first-person views of the environment and taking navigational actions. The simplest navigational agents learn a particular behaviour such as collecting or avoiding particular objects (Kempka et al., 2016; Jaderberg et al., 2016; Mirowski et al., 2016) or playing deathmatches (Lample & Chaplot, 2017; Dosovitskiy & Koltun, 2017). Subsequently, there have been efforts on training navigational agents whose behaviour is conditioned on a target specified using images (Zhu et al., 2017) or coordinates (Gupta et al., 2017a; Savva et al., 2017). More recently, there has been much interest in training agents conditioned on language as it offers several advantages over using images or coordinates.

Firstly, the compositionality of language allows generalization to new tasks without additional learning. Prior work (Oh et al., 2017; Hermann et al., 2017; Chaplot et al., 2017) has trained navigational agents to follow instructions and shown zero-shot generalization to new instructions which contain unseen composition of words seen in the training instructions. Secondly, language is also a convenient means for humans to communicate with autonomous agents. Language not only allows instruction but also interaction. Gordon et al. (2018) and Das et al. (2017) train agents to answer questions by navigating in the environment to gather the required information.

These multimodal tasks involve several challenges, such as perception from raw pixels, grounding of words in the instruction or question to visual objects and attributes, reasoning to perform relational tasks, fine-grained navigation in 3D environments with continuous state space, and learning to answer questions. Training a multi-task model can also facilitate knowledge transfer between the tasks and allow the model to generalize to scenarios which were not possible with single tasks. For example, if an agent learns to follow the instruction 'Go to the red pillar' and answer the question 'What color is the torch?', then it should also be able to follow the instruction 'Go to the red torch' and answer the question 'What color is the pillar?' without any additional training.

In this paper, we aim to train a multi-task navigation model to follow instructions and answer questions. To test the generalization of multi-task models, we define cross-task knowledge transfer as an

| Task | Train Set | Test Set |
|---|---|---|
| SGN | Instructions *not* containing 'red' & 'pillar': 
  'Go to the **blue** object' 
  'Go to the **torch**' | Instructions containing 'red' or 'pillar': 
  'Go to the red pillar' 
  'Go to the tall red object' |
| EQA | Questions *not* containing 'blue' & 'torch': 
  'Which object is red in color?' 
  'What color is the tall pillar?' | Questions containing 'blue' or 'torch': 
  'Which object is **blue** in color?' 
  'What color is the **torch**?' |

**Figure 1:** An example of first-person view in the 3D Doom environment with sample instructions and questions. The test set consists of unseen instructions and questions. The dataset evaluates a model for cross-task knowledge transfer between Semantic Goal Navigation (SGN) and Embodied Question Answering (EQA).

evaluation criteria, evaluating zero-shot learning on instructions and questions consisting of unseen composition of words in both tasks. In order to achieve cross-task knowledge transfer, words in the input space of both tasks need to be aligned with each other and with the answer space while they are being grounded to visual objects and attributes. Prior models fail to achieve this as they are designed for a single task. We propose a novel dual-attention model involving sequential Gated- and Spatial-Attention operations to perform explicit task-invariant alignment between the image representation channels and the words in the input and answer space. We create datasets and simulation scenarios for testing cross-task knowledge transfer in the Doom environment (Kempka et al., 2016) and show that the proposed model outperforms a range of baselines on both tasks. Additionally, we demonstrate that the modularity of our model allows easy addition of new objects and attributes to a trained model. We plan to open-source the implementation of our proposed model as well as the datasets and simulation environments.

## 2 RELATED WORK

This paper is motivated by a series of works on learning to follow navigation instructions (Oh et al., 2017; Hermann et al., 2017; Chaplot et al., 2017; Wu et al., 2018; Yu et al., 2018a) and learning to answer questions by navigating around the environment (Das et al., 2017; Gordon et al., 2018). Among methods learning from instructions in 3D environments, Oh et al. (2017) introduced a hierarchical RL model for learning sequences of instructions by learning skills to solve subtasks. Chaplot et al. (2017) introduced a gated-attention model for multimodal fusion of textual and visual representations using multiplicative interactions, whereas Hermann et al. (2017) introduced auxiliary tasks such as temporal autoencoding and language prediction to improve sample efficiency for this task. Yu et al. (2018a) proposed guided feature transformation which involves transformation of visual representations using latent sentence embeddings computed from the language input.

Among models for embodied question answering, Das et al. (2017) introduced a hierarchical model consisting of 4 modules, each for processing images, encoding questions, navigation, and question-answering, each of which is pretrained with supervised or imitation learning, followed by fine-tuning of the navigation model using reinforcement learning. Gordon et al. (2018) introduced the task of Interactive Question Answering which involves interacting with objects in the environment with non-navigational actions for answering questions. They proposed Hierarchical Interactive Memory Network (HIMN), which allows temporal abstraction using a factorized set of controllers.

All of the above methods are designed for a single task, following navigational instructions or answering questions, whereas we aim to train a single model for both tasks. Yu et al. (2018b) introduced a model for interactive language acquisition by training on both Visual Question Answering and following instructions in a 2D grid world environment. We aim to tackle multimodal multitask learning in challenging 3D environments. Partial observability results in the requirement of learning to navigate for answering the questions, turning visual question answering to embodied question answering. 3D environments also allow us to test on interesting and more challenging instructions based on relative size of the objects, in addition to colors and types.

In addition to the above, there is a huge body of work on multimodal learning in static settings which do not involve navigation or reinforcement learning. Some relevant works which use attention mechanisms similar to the ones used in our proposed model include Perez et al. (2017); Fukui et al. (2016); Xu & Saenko (2016); Hudson & Manning (2018); Gupta et al. (2017b) for Visual Question Answering and Zhao et al. (2018) for grounding audio to vision.

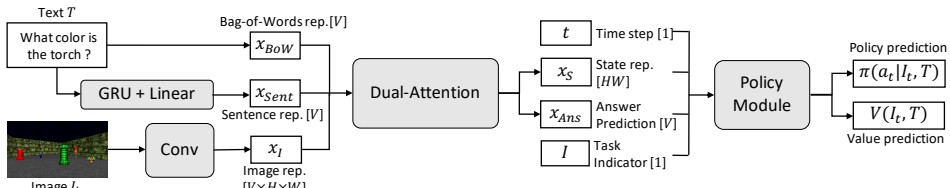

**Figure 2:** Overview of our proposed architecture, described in detail in Section 4.

## 3 PROBLEM FORMULATION

Consider an autonomous agent interacting with an episodic environment as shown in Figure 1. In the beginning of each episode, the agent receives a textual input $T$ specifying the task that it needs to achieve. For example, $T$ could be an instruction describing the target object or a question querying about some visual detail of objects in the environment. At each time step $t$, the agent observes a state $s_t = (I_t, T)$ where $I_t$ is the first-person (egocentric) view of the environment, and takes an action $a_t$, which could be a navigational action or an answer action. The agent's objective is to learn a policy $\pi(a_t|s_t)$ which leads to successful completion of the task specified by the textual input $T$.

**Tasks**. We focus on the multi-task learning of two visually-grounded language navigation tasks: In *Embodied Question Answering (EQA)*, the agent is given a question ("What color is the torch?"), and it must navigate around the 3D environment to explore the environment and gather information to answer the question ("red"). In *Semantic Goal Navigation (SGN)*, the agent is given a language instruction ("Go to the red torch") to navigate to a goal location.

**Environments**. We adapt the ViZDoom (Kempka et al., 2016)-based language grounding environment proposed by Chaplot et al. (2017) for visually-grounded multitask learning. It consists of a single room with 5 objects. The objects are randomized in each episode based on the textual input. We use two difficulty settings for the Doom domain: *Easy*: The agent is spawned at a fixed location. The candidate objects are spawned at five fixed locations along a single horizontal line in the field of view of the agent. *Hard*: The candidate objects and the agent are spawned at random locations and the objects may or may not be in the agents field of view in the initial configuration. The agent must explore the map to view all objects.

**Datasets**. We use the set of instructions from Chaplot et al. (2017) and create a dataset for questions using the same set of objects and attributes. We define cross-task knowledge transfer as an evaluation criteria for testing generalization of multi-task models. We create train-test splits for both instructions and questions datasets to explicitly test a multitask model's ability to transfer the knowledge of words across different tasks. Each instruction in the test set contains a word that is never seen in any instruction in the training set but is seen in some questions in the training set. Similarly, each question in the test set contains a word never seen in any training set question. Figure 1 illustrates the train-test split of instructions and questions used in our experiments in the Doom domain. Note that for the EQA trainset, unseen words can be present in the answer.

The agent can take 4 actions: 3 navigational actions (forward, left, right) and 1 answer action. When the agent takes the answer action, the answer with the maximum probability in the output answer distribution is used. Other details such as the train-test splits are deferred to the Appendix. We also report results on an additional environment based on House3D (Wu et al., 2018) in the Appendix.

## 4 PROPOSED METHOD

In this section, we detail our proposed architecture (illustrated in Figure 2). At the start of each episode, the agent receives a textual input $T$ (an instruction or a question) specifying the task that it needs to achieve. At each time step, the agent observes an egocentric image $I_t$ which is passed through a convolutional neural network (LeCun et al., 1995) with ReLU activations (Glorot et al., 2011) to produce the image representation $x_I = f(I_t; \theta_{\text{conv}}) \in \mathbb{R}^{V \times H \times W}$, where $\theta_{\text{conv}}$ denotes the parameters of the convolutional network, $V$ is the number of feature maps in the convolutional network output which is equal to the vocabulary size, and $H$ and $W$ are the height and width of each feature map. We use two representations for the textual input $T$: (1) the bag-of-words representation denoted by $x_{\text{BoW}} \in \mathbb{R}^V$ and (2) a sentence representation $x_{\text{sent}} = f(T; \theta_{\text{sent}}) \in \mathbb{R}^V$, which is

computed by passing the words in $T$ through a Gated Recurrent Unit (GRU) (Cho et al., 2014) network followed by a linear layer. Here, $\theta_{\text{sent}}$ denotes the parameters of the GRU network and the linear layer with ReLU activations. Next, the Dual-Attention unit $f_{\text{DA}}$ combines the image representation with the text representations to get the complete state representation $x_{\text{S}}$ and answer prediction $x_{\text{Ans}}$:

$$x_{\text{S}}, x_{\text{Ans}} = f_{\text{DA}}(x_I, x_{\text{BoW}}, x_{\text{sent}})$$

Finally, $x_S$ and $x_{\text{Ans}}$, along with a time step embedding and a task indicator variable (for whether the task is SGN or EQA), are passed to the policy module to produce an action.

## 4.1 DUAL-ATTENTION UNIT

The Dual-Attention unit uses two types of attention mechanisms, Gated-Attention $f_{\text{GA}}$ and Spatial-Attention $f_{\text{SA}}$, to align representations in different modalities and tasks.

**Gated-Attention.** The Gated-Attention unit (Figure 3) was proposed in (Chaplot et al., 2017) for multimodal fusion. Intuitively, a GA unit attends to the different channels in the image representation based on the text representation. For example, if the textual input is the instruction 'Go to the red pillar', then the GA unit can learn to attend to channels which detect red things and pillars. More specifically, the GA unit takes as input a 3-dimensional tensor image representation $y_I \in \mathbb{R}^{d \times H \times W}$ and a text representation $y_T \in \mathbb{R}^d$, and outputs a 3-dimensional ten-

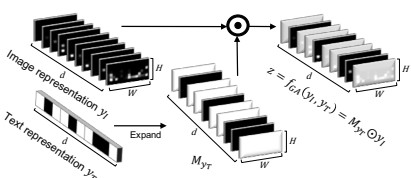

**Figure 3:** Gated-Attention unit $f_{\text{GA}}$

sor $z \in \mathbb{R}^{d \times H \times W}$. Note that the dimension of $y_T$ is equal to the number of feature maps and the size of the first dimension of $y_I$. In the Gated-Attention unit, each element of $y_T$ is expanded to a $H \times W$ matrix, resulting in a 3-dimensional tensor $M_{y_T} \in \mathbb{R}^{d \times H \times W}$, whose $(i, j, k)^{th}$ element is given by $M_{y_T}[i, j, k] = y_T[i]$. This matrix is multiplied element-wise with the image representation: $z = f_{\text{GA}}(y_I, y_T) = M_{y_T} \odot y_I$, where $\odot$ denotes the Hadamard product (Horn, 1990).

**Spatial-Attention.** We propose a Spatial-Attention unit (Figure 4) which is analogous to the Gated-Attention unit except that it attends to different *pixels* in the image representation rather than the channels. For example, if the textual input is the question 'Which object is blue in color?', then we would like to spatially attend to the parts of the image which contain a blue object in order recognize the type of the blue object. The Spatial-Attention unit takes as input a 3-dimensional tensor image representation $y_I \in \mathbb{R}^{d \times H \times W}$ and a 2-dimensional spatial attention

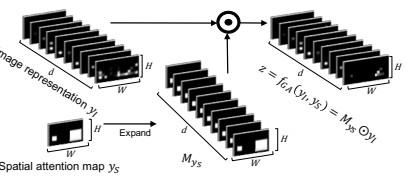

**Figure 4:** Spatial-Attention unit $f_{\text{SA}}$

map $y_S \in \mathbb{R}^{H \times W}$, and outputs a tensor $z \in \mathbb{R}^{d \times H \times W}$. Note that the height and width of the spatial attention map is equal to the height and width of the image representation. In the spatial-attention unit, each element of the spatial attention map is expanded to a $d$ dimensional vector. This again results in a 3-dimensional tensor $M_{y_S} \in \mathbb{R}^{d \times H \times W}$, whose $(i, j, k)^{th}$ element is given by: $M_{y_S}[i, j, k] = y_S[j, k]$. Just like in the Gated-Attention unit, this matrix is multiplied element-wise with the image representation: $z = f_{\text{SA}}(y_I, y_S) = M_{y_S} \odot y_I$. Similar spatial attention mechanisms have been used for Visual Question Answering (Fukui et al., 2016; Xu & Saenko, 2016; Hudson & Manning, 2018; Gupta et al., 2017b) and grounding audio in vision (Zhao et al., 2018).

**Dual-Attention**. We now describe the operations in the Dual-Attention unit shown in Figure 5, as well as motivate the intuitions behind each operation. Given $x_I$, $x_{\text{BoW}}$, and $x_{\text{sent}}$, the Dual-Attention unit first computes a Gated-Attention over $x_I$ using $x_{\text{BoW}}$:

$$x_{\text{GA1}} = f_{\text{GA}}(x_I, x_{\text{BoW}}) \in \mathbb{R}^{V \times H \times W}. \tag{1}$$

Intuitively, this first Gated-Attention unit associates each word in the vocabulary with a feature map in the image representation. A particular feature map is activated if and only if the corresponding word occurs in the textual input. In other words, the feature maps in the convolutional output learns to detect different objects and attributes, and words in the textual input specify which objects and attributes are relevant to the current task. The Gated-Attention using BoW representation attends to feature maps detecting corresponding objects and attributes, and masks all other feature maps. We

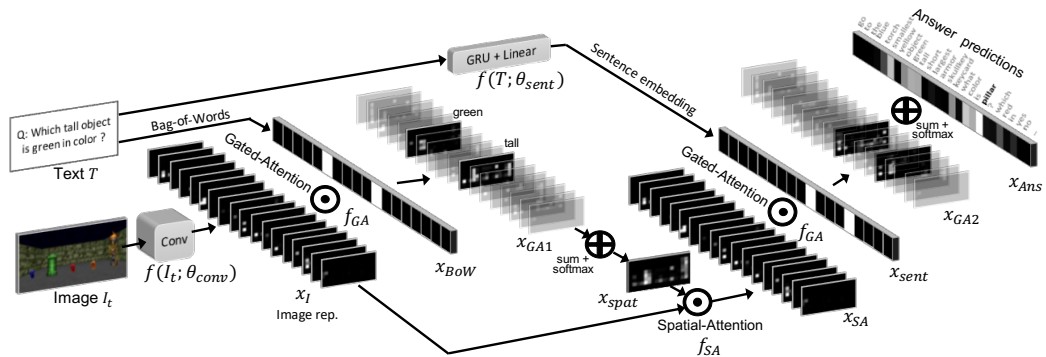

**Figure 5:** Architecture of the **Dual-Attention** unit with example intermediate representations and operations.

use bag-of-words representation for the first GA unit as it explicitly aligns the words in textual input irrespective of whether it is a question or an instruction. Note that bag-of-words representation has been used previously in models trained for learning to follow instructions (Hermann et al., 2017).

Next, the output of the Gated-Attention unit $x_{\text{GA1}}$ is converted to a spatial attention map by summing over all channels followed by a softmax over $H \times W$ elements:

$$x_{\text{spat}} = \sigma \left( \sum_i^V x_{\text{GA1}}[i, :, :] \right) \in \mathbb{R}^{H \times W} \tag{2}$$

where the softmax $\sigma(z)_j = \exp(z_j)/\sum_j \exp(z_j)$ ensures that the attention map is normalized. Summation of $x_{\text{GA1}}$ along the depth dimension gives a spatial attention map which has high activations at spatial locations where relevant objects or attributes are detected. ReLU activations in the convolutional feature maps makes all elements positive, ensuring that the summation aggregates the activations of relevant feature maps.

$x_{\text{spat}}$ and $x_I$ are then passed through a Spatial-Attention unit:

$$x_{\text{SA}} = f_{\text{SA}}(x_I, x_{\text{spat}}) \in \mathbb{R}^{V \times H \times W} \tag{3}$$

The Spatial-Attention unit outputs all attributes present at the locations where relevant objects and attributes are detected. This is especially helpful for question answering, where a single Gated-Attention may not be sufficient. For example, if the textual input is 'Which color is the pillar?', then the model needs to attend not only to feature maps detecting pillars (done by the Gated-Attention), but also to other attributes at the spatial locations where pillars are seen in order to predict their color. Note that a single Gated-Attention is sufficient for instruction following, as shown in (Chaplot et al., 2017). For example, if the textual input is 'Go to the green pillar', the first Gated-Attention unit can learn to attend to feature maps detecting green objects and pillar, and learn a navigation policy based on the spatial locations of the feature map activations.

$x_{\text{SA}}$ is then passed through another Gated-Attention unit with the sentence-level text representation:

$$x_{\text{GA2}} = f_{\text{GA}}(x_{\text{SA}}, x_{\text{sent}}) \in \mathbb{R}^{V \times H \times W} \tag{4}$$

This second Gated-Attention unit enables the model to attend to different types of attributes based on the question. For instance, if the question is asking about the color ('Which color is the pillar?'), then the model needs to attend to the feature maps corresponding to colors; or if the question is asking about the object type ('Which object is green in color?'), then the model needs to attend to the feature maps corresponding to object types. The sentence embedding $x_{\text{sent}}$ can learn to attend to multiple channels based on the textual input and mask the rest.

Next, the output is transformed to answer prediction by again doing a summation and softmax but this time summing over the height and width instead of the channels:

$$x_{\text{Ans}} = \sigma \left( \sum_{j,k}^{H,W} x_{\text{GA2}}[:, j, k] \right) \in \mathbb{R}^V \tag{5}$$

Summation of $x_{\text{GA2}}$ along each feature map aggregates the activations for relevant attributes spatially. Again, ReLU activations for sentence embedding ensure aggregation of activations for each attribute or word. The answer space is identical to the textual input space $\mathbb{R}^V$.

Finally, the Dual-Attention unit $f_{\text{DA}}$ outputs the answer prediction $x_{\text{Ans}}$ and the flattened spatial attention map $x_{\text{S}} = \text{vec}(x_{\text{spat}})$, where $\text{vec}(.)$ denotes the flattening operation.

**Policy Module**. The policy module takes as input the state representation $x_{\text{S}}$ from the Dual-Attention unit, a time step embedding $t$, and a task indicator variable $I$ (for whether the task is SGN or EQA). The inputs are concatenated then passed through a linear layer, then a recurrent GRU layer, then linear layers to estimate the policy function $\pi(a_t \mid I_t, T)$ and the value function $V(I_t, T)$.

All above operations are differentiable, making the entire architecture trainable end-to-end. Note that all attention mechanisms in the Dual-Attention unit only modulate the input image representation, i.e., mask or amplify specific feature maps or pixels. This ensures that there is an explicit alignment between the words in the textual input, the feature maps in the image representation, and the words in answer space. This forces the convolutional network to encode all the information required with respect to a certain word in the corresponding output channel. This explicit task-invariant alignment between convolutional feature maps and words in the input and answer space facilitates grounding and allows for cross-task knowledge transfer. As shown in the results later, this also makes our model modular and allows easy addition of objects and attributes to a trained model.

### 4.2 OPTIMIZATION

The entire model is trained to predict both navigational actions and answers jointly. The policy is trained using Proximal Policy Optimization (PPO) (Schulman et al., 2017). For training the answer predictions, we use a supervised cross-entropy loss. Both types of losses have common parameters as the answer prediction is essentially an intermediate representation for the policy.

**Auxiliary Task**. As mentioned earlier, the feature maps in the convolutional output are expected to detect different objects and attributes. Consequently, we add a spatial auxiliary task to detect the object or attribute in the convolutional output channels corresponding to the word in the bag-of-words representation. A prior work (Gupta et al., 2017b) also explored the use of attribute and object recognition as an auxiliary task for Visual Question Answering. Rather than doing fine-grained object detection, we keep the size of the auxiliary predictions the same as the convolutional output to avoid increase in number of parameters, and maintain the explicit alignment on the convolutional

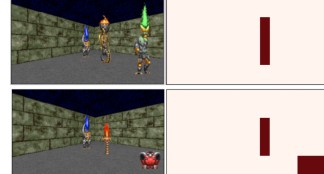

**Figure 6:** Example auxiliary task labels for the red channel.

feature maps with the words. Consequently, auxiliary labels are $(V \times H \times W)$-dimensional tensors, where each of the $V$ channels correspond to a word in the vocabulary, and each element in a channel is 1 if the corresponding object or attribute is present in the current frame spatially. Figure 6 shows examples of auxiliary task labels for the channel corresponding to the word 'red'. The auxiliary tasks are also trained with cross-entropy loss.

## 5 EXPERIMENTS & RESULTS

Jointly learning semantic goal navigation and embodied question answering essentially involves a fusion of verbal and visual modalities. While prior methods are designed for a single task, we adapt several baselines for our environment and tasks by using their multimodal fusion techniques. We use two naive baselines, **Image only** and **Text only**; two baselines based on prior semantic goal navigation models, **Concat** (used by Hermann et al. (2017); Misra et al. (2017)) and **Gated-Attention** (GA) (Chaplot et al., 2017); and two baselines based on Question Answering models, **FiLM** (Perez et al., 2017) and **PACMAN** (Das et al., 2017). For fair comparison, we replace the proposed Dual-Attention unit with multimodal fusion techniques in the baselines and keep everything else identical to the proposed model. We provide more implementation details of all baselines in the Appendix.

### 5.1 RESULTS

We train all models for 10 million frames in the *Easy* setting and 50 million frames in the *Hard* setting. We use a +1 reward for reaching the correct object in SGN episodes and predicting the

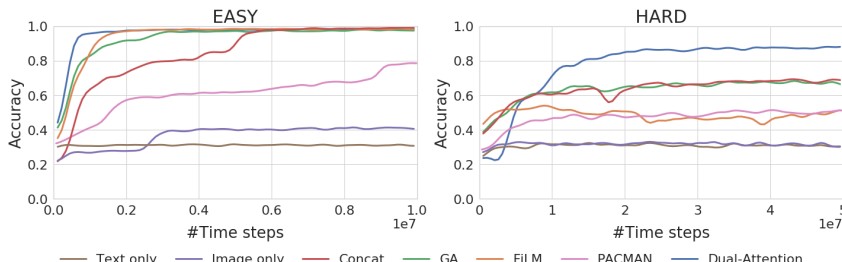

**Figure 7:** Training accuracy of all models trained with auxiliary tasks for *Easy* (left) and *Hard* (right).

**Table 1:** Accuracy of all models on SGN & EQA **test** sets for both *Easy* & *Hard* difficulties.

| | *Easy* | | | | *Hard* | | | |
| | No Aux | | Aux | | No Aux | | Aux | |
| Model | SGN | EQA | SGN | EQA | SGN | EQA | SGN | EQA |
|---|---|---|---|---|---|---|---|---|
| Text only | 0.2 | 0.33 | 0.2 | 0.33 | 0.2 | 0.33 | 0.2 | 0.33 |
| Image only | 0.20 | 0.09 | 0.21 | 0.08 | 0.16 | 0.08 | 0.15 | 0.08 |
| Concat | 0.33 | 0.21 | 0.31 | 0.19 | 0.2 | 0.26 | 0.39 | 0.22 |
| GA | 0.27 | 0.18 | 0.35 | 0.24 | 0.18 | 0.11 | 0.22 | 0.24 |
| FiLM | 0.24 | 0.11 | 0.34 | 0.12 | 0.12 | 0.03 | 0.25 | 0.15 |
| PACMAN | 0.26 | 0.12 | 0.33 | 0.10 | 0.29 | 0.33 | 0.11 | 0.27 |
| **Dual-Attention** | **0.86** | **0.53** | **0.96** | **0.58** | **0.86** | **0.38** | **0.82** | **0.59** |

correct answer in EQA episodes. We use a small negative reward of -0.001 per time step to encourage shorter paths to target and answering questions as soon as possible. We also use distance-based reward shaping for SGN episodes, where the agent receives a small reward proportional to decrease in distance to the target. In the next subsection we evaluate the performance of the proposed model without the reward shaping. SGN episodes end when the agent reaches any object, and EQA episodes when the agent predicts any answer. All episodes have a maximum length of 210 time steps. We train all models with and without the auxiliary tasks using identical reward functions.

All models are trained jointly for both the tasks and tested on each task separately. In Figure 7, we show the training performance curves for all models trained with Auxiliary tasks in both *Easy* and *Hard* settings. In Table 1, we report the test performance of all models on both SGN and EQA for both *Easy* and *Hard* settings. During training, the Dual-Attention model learns faster as compared to the baselines in the *Easy* setting while achieving higher final performance in the *Hard* setting (see Figure 7). More interestingly, as shown in Table 1, the Dual-Attention model achieves considerably higher accuracy on the test set for SGN and EQA in both the difficulty settings when trained with or without auxiliary tasks. These results confirm the hypothesis that prior models, which are designed for a single task, lack the ability align the words in both the tasks and transfer knowledge across tasks. Lower accuracy on EQA for most models (see Table 1) indicates that EQA is more challenging than SGN as it involves alignment between not only input textual and visual representations but also with the answer space. As expected, using spatial auxiliary tasks lead to better performance for all models Visualization of the attention maps and intermediate representations in the model indicate that the textual and visual representations are aligned as expected (see Appendix for visualizations)[1].

## 5.2 ABLATION TESTS

We perform a series of ablation tests in order to analyze the contribution of each component in the Dual-Attention unit: without Spatial-Attention (**w/o SA**), without the first Gated-Attention with $x_{BoW}$ (**w/o GA1**), and without the second Gated-Attention with $x_{sent}$ (**w/o GA2**). We also try removing the task indicator variable (**w/o Indicator Variable**), removing reward shaping (**w/o Reward Shaping**), and training the proposed model on a single task, SGN or EQA (**DA Single-Task**).

Figure 8 shows the training performance curves for the Dual-Attention model along with all ablation models in the *Easy* Setting. In Table 2, we report the test set performance of all ablation models. The results indicate that SA and GA1 contribute the most to the performance of the Dual-Attention

---

[1]See `https://sites.google.com/view/emml` for policy execution and visualization videos.

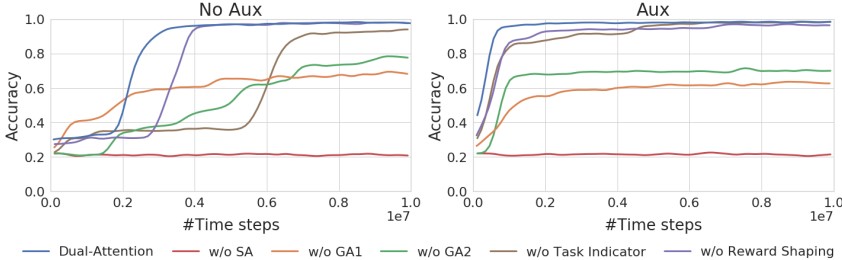

**Figure 8:** Training accuracy of proposed Dual-Attention model with all ablation models trained without (left) and with (right) auxiliary tasks for the *Easy* environment.

**Table 2:** Accuracy of all the ablation models trained with and without Auxiliary tasks on SGN and EQA test sets for the Doom Easy environment.

| Model | No Aux | | Aux | |
|---|---|---|---|---|
| | SGN | EQA | SGN | EQA |
| w/o SA | 0.20 | 0.16 | 0.20 | 0.15 |
| w/o GA1 | 0.14 | 0.25 | 0.16 | 0.38 |
| w/o GA2 | 0.80 | 0.33 | 0.97 | 0.15 |
| w/o Task Indicator | 0.79 | 0.47 | 0.96 | 0.56 |
| w/o Reward Shaping | 0.82 | 0.49 | 0.93 | 0.51 |
| DA Single-Task | 0.63 | 0.31 | 0.91 | 0.34 |
| DA Multi-Task | 0.86 | 0.53 | 0.96 | 0.58 |

**Table 3:** The performance of a trained policy appended with object detectors on instructions containing unseen words ('red' and 'pillar').

| Instruction | Easy | Hard |
|---|---|---|
| Go to the **pillar** | 1.00 | 0.71 |
| Go to the **red** object | 0.99 | 0.89 |
| Go to the tall/short **pillar** | 0.99 | 0.68 |
| Go to the <known_color>**pillar**. | 1.00 | 0.79 |
| Go to the **red** <known_object> | 1.00 | 0.93 |
| Go to the largest/smallest **red** object | 0.95 | 0.69 |
| Go to the tall/short **red pillar** | 0.99 | 0.88 |
| Go to the **red pillar** | 0.99 | 0.82 |

model. GA2 is critical for performance on EQA but not SGN (see Table 2). This is expected as GA2 is designed to attend to different objects and attributes based on the question and is used mainly for answer prediction. It is not critical for SGN as the spatial attention map consists of locations of relevant objects, which is sufficient for navigating to the correct object. Reward shaping and indicator variable help with learning speed (see Figure 8), but have little effect on the final performance (see Table 2). Dual-Attention models trained only on single tasks work well on SGN especially with auxiliary tasks. This is because the auxiliary task for single task models includes object detection labels corresponding to the words in the test set. This highlights a key advantage of the proposed model. Due to its modular and interpretable design, the model can used for transferring the policy to new objects and attributes without fine-tuning as discussed in the following subsection.

## 5.3 EXTENSION: TRANSFER TO NEW WORDS

Consider a scenario of SGN where the agent is trained to follow instructions of certain objects and attributes. Suppose that the user wants the agent to follow instructions about a new object such as 'pillar' or a new attribute such the color 'red' which are never seen in any training instruction. Prior SGN models are shown to perform well to unseen combination of object-attribute pairs (Chaplot et al., 2017), but they do not generalize well to instructions containing a new word. The model retrained only on new instructions will lead to catastrophic forgetting of previous instructions.

In contrast, our model can be used for transfer to new words by training an object detector for each new word and appending it to the image representation $x_I$. In order to test this, we train a single-task SGN model using the proposed architecture on the training set for instructions. We use auxiliary tasks but only for words in the vocabulary of the instructions training set. After training the policy, we would like the agent to follow instructions containing test words 'red' and 'pillar', which the agent has never seen or received any supervision about how this attribute or object looks visually. For transferring the policy, we assume access to two object detectors which would give object detections for 'red' and 'pillar' separately. We resize the object detections to the size of a feature map in the image representation ($H \times W$) and append them as channels to the image representation. We also append the words 'red' and 'pillar' to the bag-of-words representations in the same order such that they are aligned with the appended feature maps. We randomly initialize the embeddings of the new words for computing the sentence embedding. The results in Table 3 show that this policy generalizes well to different types of instructions with unseen words. This

suggests that a trained policy can be scaled to more objects provided the complexity of navigation remains consistent.

## 6 CONCLUSION

We proposed a Dual-Attention model for visually-grounded multitask learning which uses Gated- and Spatial-Attention to disentangle attributes in feature representations and align them with the answer space. We show that the proposed model is able to transfer the knowledge of words across tasks and outperforms the baselines on both Semantic Goal Navigation and Embodied Question Answering by a considerable margin. We showed that disentangled and interpretable representations make our model modular and allows for easy addition of new objects or attributes to a trained model. In future, the model can potentially be extended to transferring knowledge across different domains by using modular interpretable representations of objects which are domain-invariant.

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

# A  VISUALIZATIONS

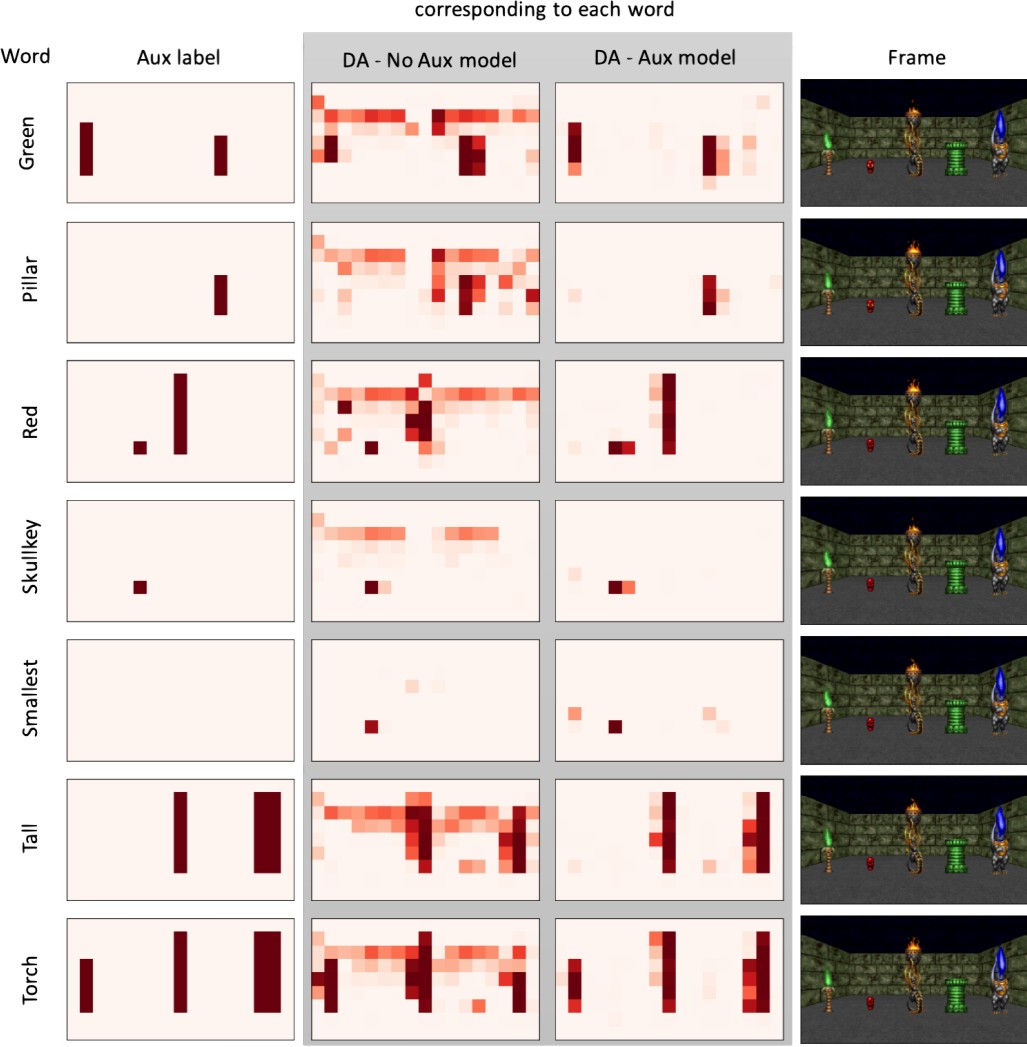

**Figure 9: Visualizations of convolutional output channels.** We visualize the convolutional channels corresponding to 7 words (one in each row) for the same frame (shown in the rightmost column). The first column shows the auxiliary task labels for reference. The second column and third column show the output of the corresponding channel for the proposed Dual-Attention model trained without and with auxiliary tasks, respectively. As expected, the Aux model outputs are very close to the auxiliary task labels. The convolutional outputs of the No Aux model show that words and objects/properties in the images have been properly aligned even when the model is not trained with any auxiliary task labels. We do not provide any auxiliary label for words 'smallest' and 'largest' as they are not properties of an object and require relative comparison of objects. The visualizations in row 5 (corresponding to 'smallest') indicate that both models are able to compare the sizes of objects and detect the smallest object in the corresponding output channel even without any aux labels for the smallest object.

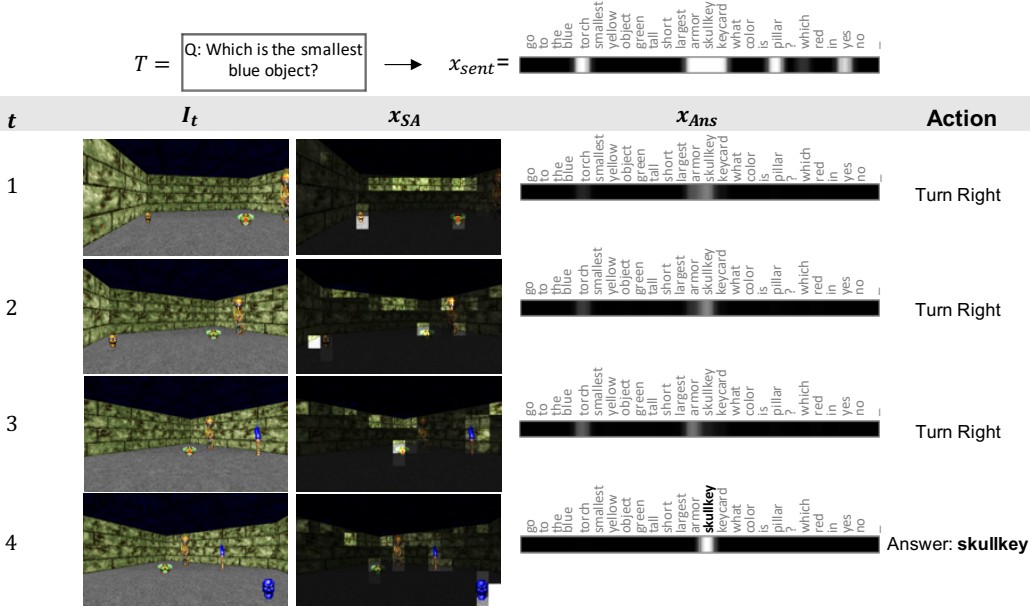

**Figure 10: Spatial Attention and Answer Prediction Visualizations.** An example EQA episode with the question "Which is the smallest blue object?". The sentence embedding of the question is shown on the top ($x_{sent}$). As expected, the embedding attends to object type words ('torch', 'pillar', 'skullkey', etc.) as the question is asking about an object type ('Which object'). The rows show increasing time steps and columns show the input frame, the input frame overlaid with the spatial attention map, the predicted answer distribution, and the action at each time step. As the agent is turning, the spatial attention attends to small and blue objects. **Time steps 1, 2**: The model is attending to the yellow skullkey but the probability of the answer is not sufficiently high, likely because the skullkey is not blue. **Time step 3**: The model cannot see the skullkey anymore so it attends to the armor which is next smallest object. Consequently, the answer prediction also predicts armor, but the policy decides not to answer due to low probability. **Time step 4**: As the agent turns more, it observes and attends to the blue skullkey. The answer prediction has high probability for skullkey as it's small and blue and the policy decides to answer the question.

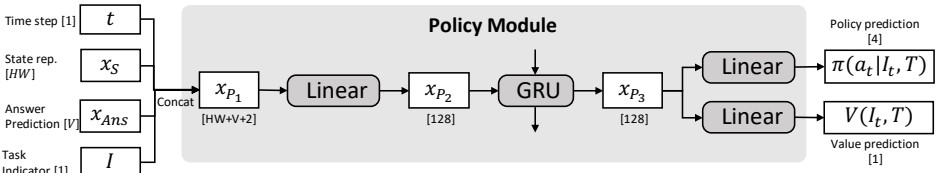

**Figure 11:** Architecture of the policy module.

# B  ADDITIONAL EXPERIMENTAL DETAILS

## B.1  HYPERPARAMETERS AND NETWORK DETAILS

The input image is rescaled to size $3 \times 168 \times 300$. The convolutional network for processing the image consisted of 3 convolutional layers: conv1 containing 32 8x8 filters with stride 4, conv2 containing 64 4x4 filters with stride 2, and conv3 containing $V$ 3x3 filters with stride 2. We use ReLU activations for conv1 and conv2 and sigmoid for conv3, as its output is used as auxiliary task predictions directly. We use word embeddings and GRU of size 32 followed by linear layer of size $V$ to get the sentence-level representation. The policy module uses hidden dimension 128 for the linear and GRU layers (see Figure 11).

For reinforcement learning, we use Proximal Policy Optimization (PPO) with 8 actors and a time horizon of 128 steps. We use a single batch with 4 PPO epochs. The clipping parameter for PPO is set to 0.2. The discount factor ($\gamma$) is 0.99. We used Adam optimizer with learning rate 2.5e-4 for all experiments.

## B.2  BASELINE DETAILS

**Image only**: Naive baseline of just using the image representation: $x_S = \text{vec}(x_I)$ where $\text{vec}(.)$ denotes the flattening operation.

**Text only**: Naive baseline of just using the textual representations: $x_S = [x_{\text{BoW}}, x_{\text{sent}}]$.

**Concat**: The image and textual representations are concatenated: $x_S = [\text{vec}(x_I), x_{\text{BoW}}, x_{\text{sent}}]$. Note that concatenation is the most common method of combining representations. Hermann et al. (2017) concatenate convolutional image and bag-of-words textual representations for SGN, whereas Misra et al. (2017) use concatenation with sentence-level textual representations.

**Gated-Attention**: Adapted from Chaplot et al. (2017), who used Gated-Attention with sentence-level textual representations for SGN: $x_S = f_{\text{GA}}(x_I, x_{\text{sent}})$.

**FiLM**: Perez et al. (2017) introduced a general-purpose conditioning method called Feature-wise Linear Modulation (FiLM) for Visual Question Answering. Using FiLM, $x_S = \gamma(x_{\text{sent}}) \odot x_I + \beta(x_{\text{sent}})$ where $\gamma(x_{\text{sent}})$ and $\beta(x_{\text{sent}})$ are learnable projections of the sentence representation.

**PACMAN**: Das et al. (2017) presented a hierarchical RL model for EQA. We adapt their method by using the attention mechanism in their QA module, which takes the last 5 frames and the text as input, and computes the similarity of the text with each frame using dot products between image and sentence-level text representations. These similarities are converted into attention weights using softmax, and the attention-weighted image features are concatenated with question embedding and passed through a softmax classifier to predict the answer distribution. For this particular baseline, we use the last 5 frames as input at each time step, unlike the proposed model and all other baselines which use a single frame as input. The attention-weighted image features are used as the state representation. The PACMAN model used a pretrained QA module, but we train this module jointly with the Navigation model for fair comparison with the proposed model.

For each of the above method except PACMAN, we use a linear layer $f$ with ReLU activations followed by softmax $\sigma$ to get a $V$-dimensional answer prediction from the state representations: $x_{\text{Ans}} = \sigma(f(x_S; \theta_{Lin}))$. $x_S$ and $x_{Ans}$ are concatenated and passed to the policy module along with the time step and task indicator variable just as in the proposed model.

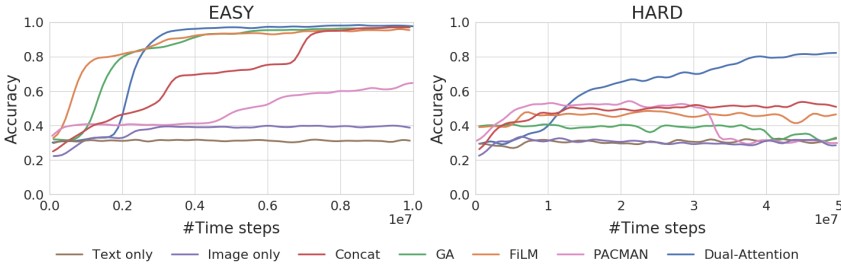

**Figure 12:** Plot showing the **training** accuracy of all the models without auxiliary tasks for Doom Easy and Hard environments.

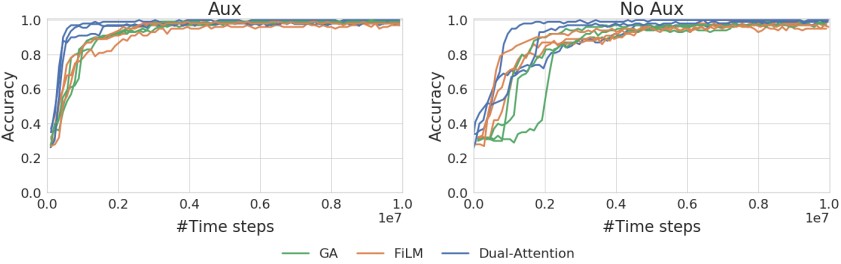

**Figure 13:** Plot showing the **training** accuracy of 3 models across 3 training runs with different seeds with and without auxiliary tasks for Doom Easy environment without any smoothing.

## C  DOOM EXPERIMENT DETAILS

**Additional Results**. We show the training accuracy of all models without auxiliary tasks for Doom Easy and Hard in Figure 12. We also show the training accuracy of 3 models across 3 training runs with different seeds with and without auxiliary tasks for Doom Easy environment in Figure 13.

**Dataset**. The Doom objects used in our experiments are illustrated in Figure 14. Instructions and questions used for training and evaluation are listed in Tables 4.

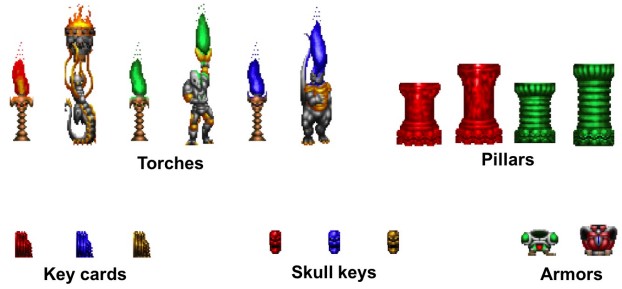

**Figure 14:** Objects of various colors and sizes used in the ViZDoom environment.

**Table 4:** Instructions and questions for ViZDoom experiments. We used 5 object classes (torch, pillar, keycard, skullkey, armor), 4 colors (red, green, blue, yellow), 2 sizes (tall, short), and 2 superlative sizes (smallest, largest).

| SGN | Instruction Type | 42 Train Instructions: Not containing 'red' & 'pillar' | 28 Test Instructions: Containing 'red' or 'pillar' |
|---|---|---|---|
| | Go to the ⟨object⟩. | torch, keycard, skullkey, armor | pillar |
| | Go to the ⟨color⟩ object. | yellow, green, blue | red |
| | Go to the ⟨size⟩ object. | tall, short | |
| | Go to the ⟨color⟩ ⟨object⟩. | blue torch, green torch, green armor, blue skullkey, blue keycard, yellow keycard, yellow skullkey | red torch, red skullkey, red pillar, green pillar, red keycard, red armor |
| | Go to the ⟨size⟩ ⟨object⟩. | short torch, tall torch | tall pillar, short pillar |
| | Go to the ⟨color⟩ ⟨size⟩ object. | green tall, blue tall, blue short, green short | red short, red tall |
| | Go to the ⟨size⟩ ⟨color⟩ object. | tall green, tall blue, short blue, short green | short red, tall red |
| | Go to the ⟨color⟩ ⟨size⟩ ⟨object⟩. | green tall torch, green short torch, blue short torch, blue tall torch | red short pillar, red short torch, red tall pillar, green tall pillar, red tall torch, green short pillar |
| | Go to the ⟨size⟩ ⟨color⟩ ⟨object⟩. | tall green torch, short green torch, short blue torch, tall blue torch | short red pillar, short red torch, tall red pillar, tall green pillar, tall red torch, short green pillar |
| | Go to the ⟨superlative⟩ object. | largest, smallest | |
| | Go to the ⟨superlative⟩ ⟨color⟩ object. | smallest yellow, smallest blue, smallest green, largest blue, largest green, largest yellow | largest red, smallest red |

| EQA | Question Type | 21 Train Questions: Not containing 'blue' & 'torch' | 8 Test Questions: Containing 'blue' or 'torch' |
|---|---|---|---|
| | What color is the ⟨object⟩? | pillar, keycard, skullkey, armor | torch |
| | What color is the ⟨size⟩ ⟨object⟩? | short pillar, tall pillar | short torch, tall torch |
| | Which object is ⟨color⟩ in color? | red, yellow, green | blue |
| | Which ⟨size⟩ object is ⟨color⟩ in color? | short red, tall red, short green, tall green | short blue, tall blue |
| | Which is the ⟨superlative⟩ object? | largest, smallest | |
| | Which is the ⟨superlative⟩ ⟨color⟩ object? | largest red, largest yellow, largest green, smallest red, smallest yellow, smallest green | largest blue, smallest blue |

## D  HOUSE3D EXPERIMENTS

In the House3D domain, we train on one house environment and randomize the colors of each object at the start of each episode. The agent's spawn location is fixed. We create instructions and questions dataset for this house similar to the Doom domain. The House3D objects used in our experiments are illustrated in Figure 15. Instructions and questions used for training and evaluation are listed in Table 6.

Each model is trained for 50 million frames jointly on both SGN and EQA, without the auxiliary tasks and using identical reward functions. Similar to Doom, we use a +1 reward for reaching the correct object in SGN episodes and predicting the correct answer in the EQA episodes. We use a small negative reward of -0.001 per time step to encourage shorter paths to target and answering the questions as soon as possible. We also use distance based reward shaping for both SGN and EQA episodes, where the agent receives a small reward proportional to decrease in distance to the target. SGN episodes end when the agent reaches any object and EQA episodes when agent predicts any answer. All episodes have a maximum length of 420 time steps.

**Table 5:** Accuracy of all the models on the SGN and EQA train and test sets for the House3D Domain.

| Model | SGN Train | SGN Test | EQA Train | EQA Test |
|---|---|---|---|---|
| Text only | 0.63 | 0.33 | 0.22 | 0.23 |
| Image only | 0.28 | 0.01 | 0.12 | 0.22 |
| Concat | 0.65 | 0.13 | 0.31 | 0.13 |
| GA | 0.98 | 0.20 | 0.92 | 0.03 |
| FiLM | 0.99 | 0.37 | 0.92 | 0.24 |
| PACMAN | 0.73 | 0.20 | 0.40 | 0.21 |
| **Dual-Attention** | **0.99** | **0.47** | **0.89** | **0.29** |

In Table 5, we report the train and test performance of all the models on both SGN and EQA. The results are similar as in Doom: the Dual-Attention model outperforms the baselines by a considerable margin.

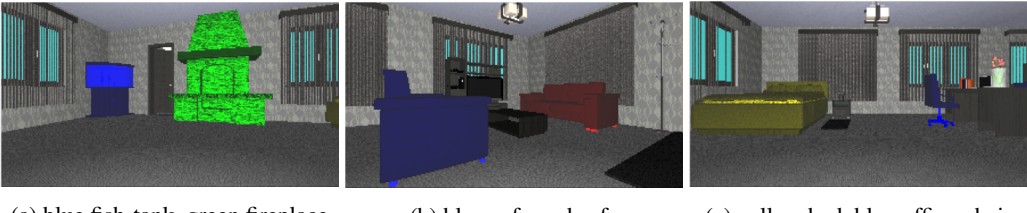

(a) blue fish_tank, green fireplace       (b) blue sofa, red sofa       (c) yellow bed, blue office_chair

**Figure 15:** Example first-person views of the House3D environment with sample objects of various colors.

**Table 6:** Instructions and questions for House3D experiments. We used 6 object classes (refrigerator, office_chair, fish_tank, fireplace, bed, sofa) and 4 colors (red, green, blue, yellow).

| SGN | Instruction Type | 22 Train Instructions: Not containing 'red' & 'bed' | 11 Test Instructions: Containing 'red' or 'bed' |
|---|---|---|---|
| | Go to the ⟨object⟩. 
 Go to the ⟨color⟩ ⟨object⟩. | refrigerator, office_chair, fish_tank, fireplace 
 green refrigerator, green office_chair, 
 green fish_tank, green fireplace, green sofa, 
 blue refrigerator, blue office_chair, 
 blue fish_tank, blue fireplace, blue sofa, 
 yellow refrigerator, yellow office_chair, 
 yellow fish_tank, yellow fireplace, yellow sofa | bed 
 red bed, green bed, blue bed, 
 yellow bed, red refrigerator, 
 red office_chair, red fish_tank, 
 red fireplace, red sofa |
| | Go to the ⟨color⟩ object. | green, blue, yellow | red |
| EQA | Question Type | 7 Train Questions: Not containing 'blue' & 'sofa' | 2 Test Questions: Containing 'blue' or 'sofa' |
| | What color is the ⟨object⟩? 
 What object is ⟨color⟩ in color? | refrigerator, office_chair, fish_tank, fireplace, bed 
 red, green, yellow | sofa 
 blue |

