# OpenReview forum: "Cross-Task Knowledge Transfer for Visually-Grounded Navigation"
_ICLR.cc/2019/Conference_

### Official Review · AnonReviewer2 · 2018-11-02
**Promising results in cross-task transfer. Missing references to prior works**

**Rating:** 5
**Confidence:** 5

**Review:**

This work proposes to train an RL-based agent to simultaneously learn Embodied Question Answering and Semantic Goal Navigation on the ViZDoom dataset. The proposed model incorporates visual attention over the input frames, and also further supervises the attention mechanism by incorporating an auxiliary task for detecting objects and attributes.

Pros:
-Paper was easy to follow and well motivated
-Design choices were extensively tested via ablation
-Results demonstrate successful transfer between SGN, EQA, and the auxiliary detection task

Cons:
-With the exception of the 2nd round of feature gating in equation (3), I fail to see how the proposed gating -> spatial attention scheme is any different from the common inner-product based spatial attention used in a large number of prior works, including  [1], [2], and [3] and many more.
-The use of attribute and object recognition as an auxiliary task for zero-shot transfer has been previously explored in [3]


Overall, while I like the results demonstrating successful inductive transfer across tasks, I did not find the ideas presented in this work to be sufficiently novel or new.

[1] Ask, Attend and Answer: Exploring Question-Guided Spatial Attention for Visual Question Answering, Huijuan Xu, Kate Saenko
[2] Drew A. Hudson, Christopher D. Manning, Compositional Attention Networks for Machine Reasoning
[3] Aligned Image-Word Representations Improve Inductive Transfer Across Vision-Language Tasks, Tanmay Gupta, Kevin Shih, Saurabh Singh, Derek Hoiem

---

> ### Author Response · Authors · 2018-11-27
> **Discussion regarding novelty, adding missing references to prior works**
>
> Thank you for providing critical feedback. We would like to address the concern of novelty in the presented work. First, irrespective of the method we believe the problem introduced in the paper is novel. It introduces not only a new non-trivial multitask learning problem of multimodal tasks with different action spaces, but also defines a novel scenario of testing cross-task knowledge transfer. We also create appropriate datasets, RL environments, and train-test splits for evaluating the above.
>
> In terms of the method, we agree that the mechanism of Gated-attention or use of auxiliary tasks is not novel. However, the purpose of the paper is not to introduce a new attention mechanism or auxiliary task, but to tackle the problem of embodied multimodal multitask learning and achieve cross-task knowledge transfer. For this purpose, we proposed a unique combination of these attention mechanisms to be used with specific kind of representations (GA-BoW followed by SA followed by GA-Sent), which we believe is novel. This unique combination leads to alignment of textual and visual representations with each other and the answer space and helps achieve cross-task knowledge transfer. We also provide justification behind the choice of this unique combination. The ablation tests indicate that these attention mechanisms by themselves or a trivial combination of the attention mechanisms are both not sufficient to achieve cross-task knowledge transfer.
>
> Thanks for pointing us to the relevant prior work in [1], [2] and [3]. We agree that the attention mechanisms in the Dual-Attention unit are similar to the attention mechanisms used in these works and we have made revisions in the paper to add these references. However, these prior works address VQA, which does not involve navigation as in EQA. In our tasks, the action space includes navigational actions as well as answer words, and we reuse the same architecture for both semantic goal navigation and EQA to enable cross-task knowledge transfer.
>
> [1] Ask, Attend and Answer: Exploring Question-Guided Spatial Attention for Visual Question Answering, Huijuan Xu, Kate Saenko
> [2] Drew A. Hudson, Christopher D. Manning, Compositional Attention Networks for Machine Reasoning
> [3] Aligned Image-Word Representations Improve Inductive Transfer Across Vision-Language Tasks, Tanmay Gupta, Kevin Shih, Saurabh Singh, Derek Hoiem

---

### Official Review · AnonReviewer1 · 2018-11-02
**Sensible model, but missing some important justification / visualization / error analysis**

**Rating:** 5
**Confidence:** 3

**Review:**

The authors propose a multitask model using a novel “dual-attention” unit for embodied question answering (EQA) and semantic goal navigation (SGN) in the virtual game environment ViZDoom. They outperform a number of baseline models originally developed for EQA and SGN (but trained and evaluated in this multitask paradigm).

Comments and questions on the model and evaluation follow.

1. Zero-shot transfer claim:
1a. This is not really zero-shot transfer, is it? You need to derive object detectors for the meanings of the novel words (“red” and “pillar” from the example in the paper). It seems like this behavior is supported directly in the structure of the model, which is great — but I don’t think it can be called “zero-shot” inference. Let me know if I’ve misunderstood!
1b. Why is this evaluated only for SGN and not for EQA?

2. Dual attention module:
2a. The gated attention model only makes sense for inputs in which objects or properties (the things picked out by convolutional filters) are cued by single words. Are there examples in the dataset where this constraint hold (e.g. negated properties like “not red”)? How does the model do? (How do you expect to scale this model to more naturalistic datasets with this strong constraint?)
2b. A critical claim of the paper is that the model learns to “align the words in both the tasks and transfer knowledge across tasks.” (Earlier in the paper, the claim is that “This forces the convolutional network to encode all the information required with respect to a certain word in the corresponding output channel.”) I was expecting you would show some gated-attention visualizations (not spatial-attention visualizations, which are downstream) to back up this claim. Can you show me visualizations of the gated-attention weights (especially when trained on the No-Aux task) which demonstrate that words and objects/properties in the images have been properly aligned? Show that e.g. the filter at index i only picks out objects/properties cued by word i?

3. Auxiliary objective: it seems like this objective solves most of the language understanding problem relevant in this task. Can you motivate why it is necessary? What is missing in the No-Aux condition, exactly? Is it just an issue with PPO optimization? Can you do error analysis on No-Aux to motivate the use of the Aux task?

4. Minor notes:
4a. In appendix A, the action is labeled “Turn Left” but the frames seem to suggest that the agent is turning right.
4b. How are the shaded regions estimated in figs. 7, 8? They are barely visible — are your models indeed that consistent across training runs? (This isn’t what I’d expect from an RL model! This is true even for No-Aux..?)
4c. Can you make it clear (via bolding or coloring, perhaps) which words are out-of-vocabulary in Table 3? (I assume “largest” and “smallest” aren’t OOV, for example?)

---

> ### Author Response · Authors · 2018-11-27
> **New visualization figures and videos showing alignment of words with corresponding convolutional output channels, discussion regarding need for auxiliary task and use of single word attributes**
>
> Thank you for providing critical feedback.
>
> 1a. Regarding the zero-shot transfer claim:
> We used the term 'zero-shot' to imply that we do not use any trajectories for training the policy after transfer. So it is zero-shot transfer for the policy, but we agree that some static samples are required for training the object detectors. We have revised the manuscript to remove the claim of zero-shot transfer.
>
> 1b. “Why is this evaluated only for SGN and not for EQA?”:
> This is because we need additional information for EQA (e.g., whether the new word corresponds to an object type or attribute), which cannot be obtained from an object/attribute detector.
>
> 2a. Use of single word objects and attributes:
> We agree that the proposed model only handles objects and attributes described by single words. One possible way to scale the proposed architecture to more naturalistic datasets with multi-word objects and attributes is to replace the Bag-of-Words representation by a ‘Bag-of-Concepts’ representation where a separate module is trained to extract ‘concepts’ from language input. ‘Skull key’ or ‘metallic gray’ can be classified as concepts. The gated-attention can then work on concepts rather than words.
> While our model might be able to handle multi-word object names and attributes using a ‘Bag-of-Concepts’ representation, more complex architectures are certainly required to handle other complexities of language such as negation, conjunctions, prepositions and so on. In future, we plan to look into learning to compose dual-attention units in a recursive manner to capture the relationships between words and phrases in order to handle these complexities. The fact that all the baselines fail to generalize better than a naive Text-only model emphasizes the difficulty of the task even with single word objects and attributes.
>
> 2b. Visualization of convolutional outputs:
> Thanks for bringing up this interesting point. We added visualizations of the convolutional network outputs in Figure 9 in Appendix A for both Aux and No Aux models. We visualize the output corresponding to 7 words for the same frame along with the auxiliary task labels. As expected, the Aux model predictions are very close to the auxiliary task labels. More interestingly, the convolutional outputs of the No Aux model show that words and objects/properties in the images have been properly aligned even when the model is not trained with any auxiliary task labels. We also uploaded some visualization videos showing convolutional output channels corresponding to different words as well as spatial attention at the following link:
> https://sites.google.com/view/emml
>
> 3. Need for auxiliary objectives:
> The most important benefit of using auxiliary tasks is the improvement in sample efficiency. Although the No Aux model can perform as well as Aux model given enough training data, the sample efficiency of Aux models is much better than No Aux models as shown by the training curves in Figure 7 (Aux) and Figure 12 (No Aux) in the appendix. The only way of learning new objects without auxiliary tasks is randomly bumping into objects or predicting an answer and receiving rewards and making a connection between the visual and textual modalities and the rewards. Auxiliary tasks provide additional supervision which helps in training the model much faster. As we scale embodied RL agents to learn thousands of objects and attributes, it might not be possible to train such models without auxiliary tasks as the trial-and-error method of learning new objects and attributes might require samples beyond the current computing capabilities. The sample inefficiency is partly an optimization issue (not just PPO but any RL algorithm in general) but also the task itself is very challenging with just sparse rewards.
>
> While auxiliary tasks improve the sample efficiency, we show that the proposed model works well even without the auxiliary tasks. This is useful for scenarios where we do not have the auxiliary labels for all or a subset of the objects and attributes.
>
> Response to minor notes
> 4a. We revised Figure 10 to correctly reflect the actions.
> 4b. The shaded region does not represent the variance across multiple runs. Each point in the curves in fig 7 and 8 is calculated using weighted running average across time (Gaussian smoothing), the shaded region represents the amount of smoothing applied at each point in a single run. We removed the shaded region in all relevant figures to avoid this confusion. We also added training curves for 3 top models across 3 training runs with different seeds both with and without auxiliary tasks for Easy environment in Figure 13 in the appendix C. The performance of our proposed model is consistent across different runs. The variance across different runs is higher in the NoAux setting as compared to the Aux setting.
> 4c. The out of vocabulary words are 'red' and 'pillar' as mentioned in the text. We made revisions to specify this in Table 3 as well.

---

> > ### Comment · AnonReviewer1 · 2018-12-08
> > **Response to authors**
> >
> > Hi authors — thanks for your answers and updates to the paper. While the gated attention mechanism designed in this paper seems to yield nice interpretable representations (thanks for Fig 9!), I still can't see how this gating mechanism can scale to anything like natural language — take the more complex sentences in the Embodied QA dataset [1], for example — without major revisions. As such, it's not clear why we should take the Dual GA mechanism to be an important milestone on the way to bigger EQA accomplishments.
> >
> > Re #3: No-Aux seems to converge on this task, true — but the representations in figure 9 suggest that convergence is nowhere near optimal. I'm worried the model with the No-Aux constraint will also not scale as-is.
> > I understand that, while learning in this task with model-free RL, there is little an agent can do to learn a word but to accidentally bump into the relevant object or accidentally predict the correct answer. This seems like a fundamental direction for improvement, either by model iteration or novel task design, since *human children* clearly learn much more than the model-free exploration in the current task can support.
> >
> > Thanks for adding figure 13 — but please note in the caption that these are not an arbitrary "3 models", but the 3 *best* models. (How many training trajectories look like these 3 models? How many don't converge?)
> >
> > Due to the above concerns about model scalability and the framing of the task, I'm keeping my rating as-is.
> >
> > [1]: https://embodiedqa.org/

---

### Official Review · AnonReviewer3 · 2018-11-05
**The manuscript is clearly written and  adds to the state of the art in multidomain machine learning.  Recommend as a poster.**

**Rating:** 7
**Confidence:** 4

**Review:**

The system is explained thoroughly, and with the help of nice looking graphics the network architecture and its function is clearly described. The paper validates the results against baselines and shows clearly the benefit of double  domain learning. The paper is carefully written and  follows the steps required for good scientific work.

Personally, I do not find this particularly original, even with the addition of the zero-shot learning component.

As a side note, the task here does not seem to need a multitask solution. Adding the text input as subtitles to the video gives essentially the same information that is used in the setup. The resulting inclusion of text could utilise the image attention models in a similar manner as the GRU is used in the manuscript for the text. In this case the problem stated in the could be mapped  to a "DeepMind Atari" type of RL solution, with text as a natural component, but added as visual clue to the game play. Hence, I am not convinced that the dual attention unit is essential to the performance the system.

In addition, there are studies (https://arxiv.org/abs/1804.03160) where sound and video are , in unsupervised manner, correlated together. This contains analogous dual attention structure as the manuscript describes, but without reinforcement learning component.

I would recommend this as a poster.

---

> ### Author Response · Authors · 2018-11-27
> **Regarding the need for multimodal learning and dual-attention**
>
> Thank you for providing critical feedback.
>
> Regarding the need for multimodal learning and dual-attention:
> It is true that the visual and textual modalities can be fused by using subtitles. However, the alignment of knowledge between the words in the subtitle, visual objects, and the answer space still remains challenging even in this setting. We believe that the dual attention unit is still essential in this setting, as a "DeepMind Atari" type of RL solution would still overfit to the training set of instructions and questions (similar to all the baselines used in our paper), and would not generalize to instructions or questions with unseen words. To verify this, we ran an experiment, where we superimposed the text in the top part of the image and trained a unimodal RL network with the same convolutional network and recurrent policy module as in our proposed model using PPO. The test performance of the model was similar to the Image only baseline (0.21 SGN accuracy, and 0.10 EQA accuracy for Easy Aux setting). We believe that this modified unimodal setting involves all the challenges of our setting and an additional challenge of learning to extract text from the image.
>
> “In addition, there are studies (https://arxiv.org/abs/1804.03160) where sound and video are , in unsupervised manner, correlated together.”
>
> Thank you for pointing us to this relevant work. We have made revisions to discuss this work with respect to our model.

---

### Author Response · Authors · 2018-11-29
**List of revisions, looking forward to further discussion**

For the convenience of the reviewers and the Area Chair, we would like to list the revisions made to the submission after the reviews:

- Addition of visualization figures and videos:
As requested by Reviewer 1, we added policy execution videos with visualization of convolutional output channels and spatial attention at the following link:
https://sites.google.com/view/emml
We also added Figure 9 containing visualization of the convolutional outputs channels corresponding to different words. Both the videos and figure indicate that our proposed model learns to detect objects and attributes corresponding to the word in relevant convolutional output channels even when it is not trained with auxiliary labels.

- Addition of training curves with different seeds:
As requested by Reviewer 1, we added training curves for top 3 models across 3 training runs with different seeds both with and without auxiliary tasks for Easy environment in Figure 13 in the appendix C. The performance of our proposed model is fairly consistent across different runs. The variance across different runs is higher in the NoAux setting as compared to the Aux setting. We also removed the shaded region in other training curves to avoid confusion.

- Addition of references to missing prior work:
In the original submission, we discussed prior work on visually-grounded navigation tasks. We thank Reviewers 2 and 3 for pointing us to relevant prior work on multimodal learning in static settings which do not involve navigation or reinforcement learning. We added references to prior methods which use similar attention mechanisms for Visual Question Answering [1,2,3] and grounding audio to vision [4]. We also added a reference to prior work which explored the use of object recognition as an auxiliary task for Visual Question Answering [3]. In contrast to these works, we propose a new non-trivial multitask learning problem of multimodal tasks with different action spaces, and define a novel scenario of testing cross-task knowledge transfer. We also create appropriate datasets, RL environments, and train-test splits for evaluating the above. Even though the individual attention mechanisms are similar to ones used in prior work, we propose a unique combination of these attention mechanisms to be used with specific kind of representations in order to achieve cross-task knowledge transfer.

- Additional experiment on using text as subtitles:
As requested by Reviewer 3, we ran an experiment where we superimposed the text in the top part of the image and trained a unimodal RL network with the same convolutional network and recurrent policy module as in our proposed model using PPO. The test performance of the model was similar to the Image only baseline (0.21 SGN accuracy, and 0.10 EQA accuracy for Easy Aux setting). This highlights the need for multimodal learning and dual-attention.

- Minor changes:
Removed the claim of zero-shot transfer in the extension to new words using object detectors.
Revised Figure 10 to correctly reflect the actions.
Revised Table 3 to specify out of vocabulary words.

If there are any follow-up questions or additional queries, we will be happy to provide additional details. We look forward to actively participate in the follow-up discussion.

[1] Ask, Attend and Answer: Exploring Question-Guided Spatial Attention for Visual Question Answering, Huijuan Xu, Kate Saenko
[2] Compositional Attention Networks for Machine Reasoning. Drew A. Hudson, Christopher D. Manning
[3] Aligned Image-Word Representations Improve Inductive Transfer Across Vision-Language Tasks, Tanmay Gupta, Kevin Shih, Saurabh Singh, Derek Hoiem
[4] The Sound of Pixels. Hang Zhao, Chuang Gan, Andrew Rouditchenko, Carl Vondrick, Josh McDermott, Antonio Torralba

---

### Meta-Review · Area_Chair1 · 2018-12-16

**Confidence:** 4
**Recommendation:** Reject

**Metareview:**

The authors have proposed a language+vision 'dual' attention architecture, trained in a multitask setting across SGN and EQA in vizDoom, to allow for knowledge grounding. The paper is interesting to read. The complex architecture is very clearly described and motivated, and the knowledge grounding problem is ambitious and relevant. However, the actual proposed solution does not make a novel contribution and the reviewers were unconvinced that the approach would be at all scalable to natural language or more complex tasks. In addition, the question was raised as to whether the 'knowledge grounding' claims by the authors are actually much more shallow associations of color and shape that are beneficial in cluttered environments.
This is a borderline case, but the AC agrees that the paper falls a bit short of its goals.